# SomaVR: A low-cost virtual reality platform and implementation framework for medical education in resource-limited settings

Mike Nsubuga [1,2,3*], Grace Kebirungi[1,2], Helen Please[4,5], Paul Buyego[2], Henry Mutegeki[1,2], Rodgers Kimera[1], Jag Dhanda[6,7], Phil Cruz[8], Meghan McCarthy[8], Darrell Hurt [8], Maria Y. Giovanni[8], Christopher Whalen[8], Michael Tartakovsky[8], Daudi Jjingo [1,2,9*]

1 The African Center of Excellence in Bioinformatics and Data-intensive Sciences, Makerere University, Kampala, Uganda, 2 Infectious Diseases Institute, Makerere University, Kampala, Uganda, 3 Faculty of Health Sciences, University of Bristol, Bristol, United Kingdom 4 Oxford University Hospitals NHS Foundation Trust, Oxford, United Kingdom, 5 Harris Manchester College, University of Oxford, Oxford, United Kingdom, 6 Brighton and Sussex Medical School, Brighton, United Kingdom, 7 Queen Victoria Hospital NHS Foundation Trust, East Grinstead, United Kingdom, 8 Office of Cyber Infrastructure and Computational Biology, National Institute of Allergy and Infectious Diseases, National Institutes of Health, Bethesda, Maryland, United States of America, 9 Department of Computer Science, College of Computing and Information Sciences, Makerere University, Kampala, Uganda

* daudi.jjingo@mak.ac.ug (DJ); nsubugamike021@gmail.com (MN)

## Abstract

Quality medical training is vital for effective healthcare worldwide. In low- and middle-income countries (LMICs), traditional training methods often face significant challenges, including limited resources, logistical barriers, and difficulties in safely replicating high-risk scenarios for infectious diseases like COVID-19 and Ebola. Additionally, medical training demands high costs, significant time, and specialized supervision, limiting its accessibility. Although virtual reality (VR) offers promising solutions to these problems, most evidence comes from high-income settings, leaving limited guidance on implementation in resource-constrained settings. We developed SomaVR, a low-cost VR platform and implementation framework for medical training in LMICs. Built with Unity3D, 'SomaVR' (soma - Swahili/Luganda for "to learn") integrates 360-degree and interactive virtual environments to create customizable training experiences aligned with specific curricula needs. Beyond the software, the framework provides a structured approach covering hardware selection, software architecture, content development workflows, and strategies for local capacity building. The platform prioritizes cross-platform compatibility, offline functionality, and cost-effective deployment. SomaVR's modular components support both high-end VR systems and low-cost solutions such as smartphone-based. The platform and framework were validated through two independent case studies: 1. COVID-19 infection prevention; and 2. Surgical training. In the surgical training, trainers from a high-income country guided Ugandan learners remotely, illustrating SomaVR's

**Data availability statement:** The source code for the VR platform is provided on the GitHub page: https://github.com/aceuganda/somavr.

**Funding:** This work was funded by Makerere University under the Research and Innovation Fund(to DJ, GK, PB, HM, MN). Hardware equipment and technical support were facilitated by the U.S. National Institute of Allergy and Infectious Diseases (NIAID) under the Office of Cyber Infrastructure and Computational Biology(to RK, PC, MM, DH, MYM, CW, MT) as part of the African Centers of Excellence in Bioinformatics and Data Intensive Sciences program. The funders had no role in study design, data collection and analysis, decision to publish, or preparation of the manuscript.

**Competing interests:** The authors have declared that no competing interests exist.

potential for long-distance knowledge exchange. In both cases, cohorts trained using SomaVR consistently outperformed those receiving conventional training, with significant improvements in procedural understanding and user engagement. Our findings also highlight that as VR technology costs decline, frugal approaches such as delivering 360-degree video via smartphone can maintain educational effectiveness in low-resource environments. This paper provides a practical blueprint for developing and implementing sustainable VR medical training platforms in resource-limited settings. By detailing the technical framework, development processes, and implementation strategies of SomaVR, we offer a replicable model for institutions seeking to leverage VR technology for medical education in LMICs.

## Author summary

In our work, we developed a low-cost virtual reality system that helps improve medical training in areas with limited resources. We built a tool that combines immersive 360-degree videos with interactive simulated environments to recreate real-life medical scenarios safely. Our goal was to overcome challenges such as high training costs, scarce equipment, and the difficulty of practicing in high-risk situations like disease outbreaks or surgical procedures. We tested our system in two different settings: one focused on preventing infections during the COVID-19 pandemic and another aimed at enhancing surgical training. In both cases, we found that our approach not only improved learning outcomes but also increased engagement compared to traditional training methods. We believe that our framework offers a practical and affordable way to deliver high-quality medical training in environments where resources are scarce. Our experience shows that it is possible to build and maintain a sustainable training system locally, and we hope that others can adapt our model to improve healthcare education worldwide.

## 1. Introduction

In the rapidly evolving landscape of global health, the effectiveness of medical training is a critical determinant of healthcare quality and accessibility. This is particularly true in low- and middle-income countries (LMICs), where healthcare systems face chronic challenges such as resource limitations, uneven distribution of skilled healthcare workers, and infrastructural deficiencies [1–3], These challenges are compounded by the lack of continuing professional development and medical education programs which are vital for maintaining healthcare standards and addressing emerging health threats [4].

Traditional medical training methods in these settings are often constrained by physical and financial resource limitations, as well as logistical complexities of reaching geographically dispersed populations [5]. The onset of the COVID-19 pandemic

further highlighted the risks and limitations of conventional face-to-face training methods, particularly in contexts where infectious diseases posed a significant threat to both healthcare providers and patients [6]. This situation was exacerbated in LMICs where protective measures and infrastructure were often insufficient to prevent the spread of infections during training sessions [1,6,7]. Additionally, the need for rapid dissemination of new medical knowledge and ongoing professional development during the pandemic underscored the necessity for flexible and innovative solutions like e-learning and immersive technologies [5].

Virtual Reality (VR) has emerged as a transformative technology for addressing the above issues. Its ability to simulate complex medical scenarios in controlled interactive environments makes it an unparalleled tool for medical education [8]. VR provides immersive and engaging learning experiences which have been shown to enhance procedural training. In surgical education, VR has demonstrated improvements in procedural times, task completion, and accuracy, alongside high user satisfaction and cost-effectiveness [9,10]. Furthermore, VR enables the replication of rare, complex, or resource-intensive scenarios without the associated risks or costs, making it particularly valuable in resource-constrained settings [11].

Despite these advantages, a recent review by Mergen et al [12] highlights several limitations. The variability and completeness of complex training scenarios can affect the reliability of learning outcomes, particularly when simulating high-stakes or nuanced medical procedures. Challenges also remain in accurately representing real-life scenarios with realistic human behavior and feedback. Financial barriers persist due to the need for high-performance equipment, interdisciplinary development teams, and ongoing technical support, although hardware costs are decreasing. These limitations emphasize the need for VR solutions that are cost-effective, technically robust, and designed with the input of end-users and medical educators [12].

In recent years, a few initiatives have explored the use of VR for healthcare training in LMICs. For instance, the LIFE:VR project in Kenya demonstrated the feasibility of VR-based emergency training but noted that high equipment costs and technical requirements limited scalability [13]. Similarly, a recent systematic review of AR and VR technologies in LMIC medical education reported that while VR shows strong educational potential, implementation remains hindered by connectivity issues, reliance on external technical support, and limited local development capacity [14]. These experiences highlight the need for adaptable and contextually grounded frameworks that address such constraints.

This study focuses on the development and implementation of SomaVR, soma means "to learn" in both Luganda (the most commonly spoken language in Uganda) and Swahili (an official language of Uganda, Kenya and Tanzania). The aim of developing SomaVR was to design a low-cost VR platform to address the educational challenges faced by healthcare professionals in Uganda by leveraging both interactive VR simulations and immersive 360-degree video environments to enhance medical training.

SomaVR was evaluated through two targeted and independent studies. The first assessed its feasibility and effectiveness as a training tool for infection prevention and control (IPC) during the COVID-19 pandemic among healthcare workers in Uganda [7]. The second examined its use in a surgical training course, where instructors based in a high-income country (HIC), in this case the United Kingdom [15], demonstrated essential surgical skills to doctors and medical students in Uganda. Both studies demonstrated SomaVR's potential to improve learning outcomes, long-distance knowledge exchange, and cross-cultural knowledge exchange, highlighting its role as an innovative solution to persistent gaps in medical education in resource-limited settings.

While our previous publications have reported on the feasibility and educational outcomes of two distinct VR-based training interventions [7,15], they did not provide a detailed account of the underlying technical systems or the implementation framework that enabled them. This manuscript fills that gap by providing the first comprehensive technical description of the SomaVR platform. Our goal is to offer a practical, replicable blueprint covering software architecture, content development workflows, and deployment strategies for other institutions, particularly those in LMICs, seeking to build and sustain their own VR-based medical training programs.

## 2. Methods

### 2.1 Team composition

The COVID-19 IPC study involved a core development team of approximately 12 individuals: 2 videographers, 1 Unity3D developer, 3 software developers (including interns from our capacity-building program), 4 clinicians, 1 educational specialist, and 1 project coordinator. Content development spanned approximately 4 months, with an estimated total of 800–1000 person-hours for filming, software development, testing, and deployment.

The surgical training study required a smaller core team of approximately 8 individuals: 2 videographers (UK-based), 1 technical coordinator, 4 surgeons (content experts), and 1 project coordinator. Given the use of minimally-processed live 360-degree video rather than complex interactive modules, development time was compressed to approximately 1 month with an estimated total of 200–300 person-hours. However, the live delivery format required real-time technical support during the week-long course.

### 2.2 Content development and curriculum design

In both studies there was extensive medical curriculum design carried out by the clinical team and educational experts, from which it was determined the appropriate form of VR for each content segment. Discussions between the medical and technical teams guided the decision-making process, to ensure technological feasibility of the curriculum while maintaining educational standards. For broader accessibility, especially in regions with limited internet connectivity or advanced hardware, most content was delivered via 360-degree videos. This method allowed learners to access materials on their personal smartphones, with potential to download them for offline use (this was demonstrated in the COVID-19 study). For the surgical training study, content was delivered as live-streamed demonstrations, enabling real-time viewing by participants in Kampala and remotely. This live delivery approach was chosen to facilitate immediate knowledge transfer from UK-based surgical instructors to Ugandan learners. Recordings were subsequently made available for review, allowing participants to revisit procedures. Content that required more detailed interaction was developed for interactive VR, accessible at equipped facilities like the ACE lab at Makerere. This complex process of VR-enabled course development is summarized in the SomaVR framework (Fig 1).

### 2.3 Collaborations

The development and implementation of SomaVR were made possible through a multi-institutional collaboration, integrating expertise from different domains. The National Institute of Allergy and Infectious Diseases (NIAID) Visualization Lab contributed to the technical aspects of VR equipment, while VRiMs and Enduvo supported interactive module creation. IDI-Makerere led the conceptualization and coordination, ensuring the framework aligned with local medical education needs. Research Education Network Uganda (RENU) facilitated enhanced internet bandwidth for the surgical training module, while additional support from ACE and the Makerere Research and Innovation Fund enabled broader development and deployment. This collaborative model highlights how a modular, accessible VR framework like SomaVR can be adapted and expanded through partnerships with institutions that bring diverse technical and educational expertise.

### 2.4 Internet requirements

The internet requirements for SomaVR varied depending on the course content and delivery mode. The COVID-19 IPC training was fully accessible offline, as all 360-degree video content was preloaded onto devices, eliminating the need for an internet connection. This ensured the deployment in settings with limited or no connectivity. In contrast, the surgical training study relied on live-streaming content, necessitating a stable and high-bandwidth internet connection. To accommodate this, we collaborated with the RENU internet service provider [16], a not-for-profit National Research Education Network (NREN) which increased the available bandwidth to 1 Gbps, ensuring smooth content delivery. While internet

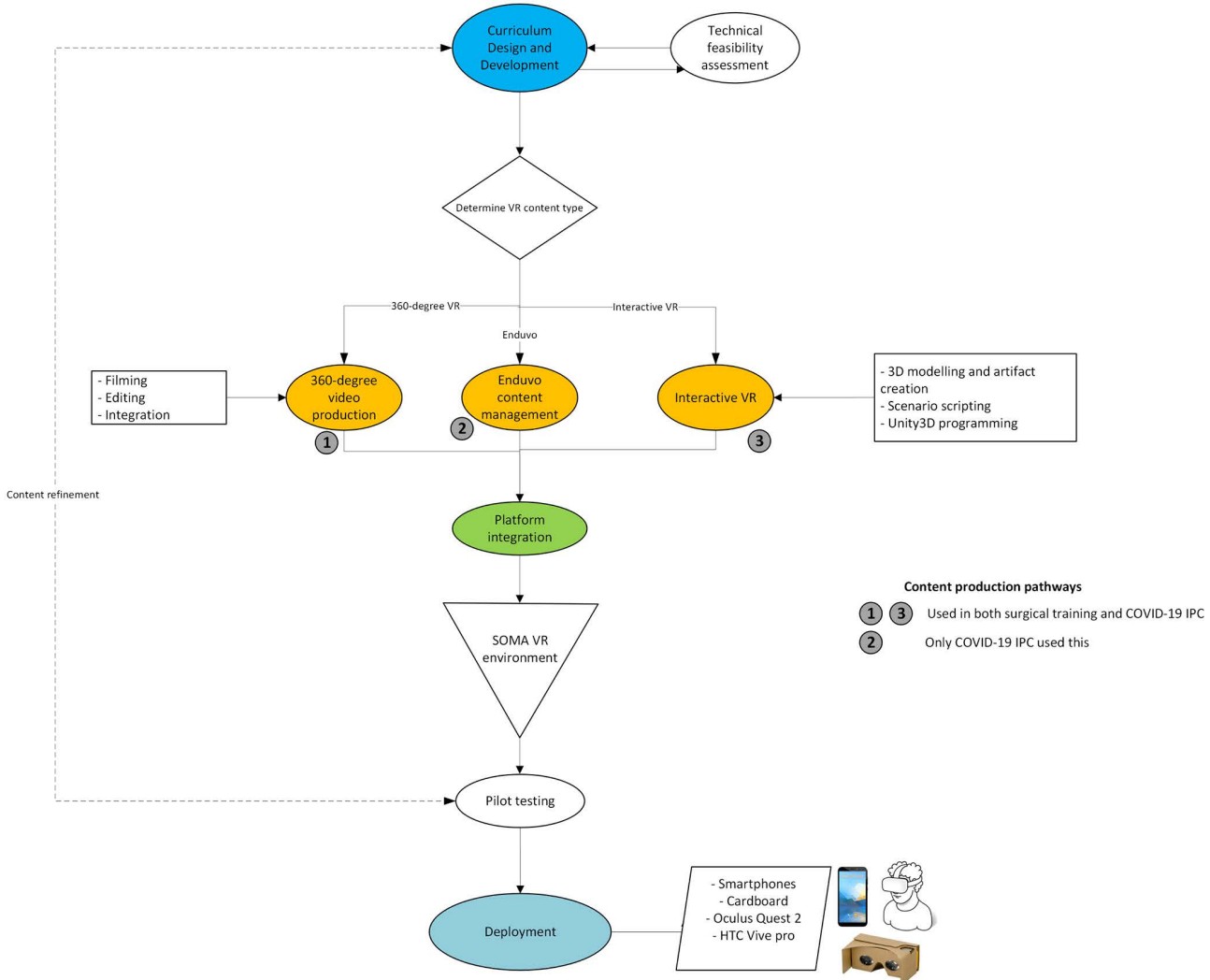

**Fig 1. The SomaVR framework: Workflow of the content development and deployment process.** This diagram illustrates the comprehensive process of designing, developing, and deploying the SomaVR educational platform. It highlights the steps from curriculum design through technical feasibility assessment (Blue), content production in both 360-degree VR and interactive VR formats (Yellow), and final integration into the SomaVR environment for deployment across various devices (Green). COVID-19 IPC utilized all three content production pathways (360-degree video production, Enduvo content management, and Interactive VR), whereas the Surgical Training module used only 360-degree video production and Interactive VR.

coverage via mobile networks is increasingly ubiquitous in Uganda; challenges remain, particularly in areas with limited fiber or broadband infrastructure [17].

**2.4.1 Technical specifications.** The technical setup varied between the two studies according to their delivery modes. In the COVID-19 IPC training, all 360-degree video modules were preloaded onto Oculus Quest 2 and HTC Vive Pro headsets to enable full offline access. Each 5-minute 4K video averaged 700–900 MB, while interactive modules created in Enduvo ranged between 50 and 100 MB. This approach removed the need for internet connectivity during training and ensured feasibility in low-connectivity settings.

The surgical training study, which relied on live-streamed 4K 360-degree video at 30 fps, required a stable high-bandwidth connection. Bandwidth at the ACE laboratory was upgraded to 1 Gbps through collaboration with RENU,

supporting up to 30 concurrent users. To maintain high-definition streaming quality, a bandwidth of 15–25 Mbps per user was recommended, with a minimum of 10 Mbps for remote participants using personal devices.

## 2.5 Hardware devices used

This study utilized a range of VR hardware and filming techniques to create immersive training environments. The selection of hardware was intentionally diverse to ensure the SomaVR framework's adaptability to the varied technological landscapes present in VR and LMICs. Videos were recorded using a GoPro MAX 360-degree camera (https://gopro.com/), which captures every angle of the environment using 3 inbuilt cameras simultaneously, providing a complete panoramic view. This setup was crucial for the development of 360-degree video content, designed to give learners the sense of being physically present in the training scenarios. For the COVID-19 IPC study the filming was carried out in Mulago Hospital, Kampala, Uganda, led by the African Center of Excellence in Bioinformatics and Data Intensive Sciences (ACE) team and Ugandan clinicians. This was delivered as pre-recorded content integrated by Enduvo into immersive VR content delivered as part of a 2-week course at ACE, Infectious Diseases Institute Makerere (IDI). For the surgical training study, the filming was carried out in Brighton and Sussex Medical School, Brighton, UK, led by the Virtual Reality in Medicine and Surgery (VRiMS) team and UK clinicians. This was delivered as live content as part of a week-long course at ACE, IDI and remotely to participants in 25 countries. While an overview of these studies is presented here, detailed descriptions of the curriculum design, content, and evaluation are available in the original publications: Buyego et al. (2022) [7]for the COVID-19 Infection Prevention Control(IPC) study and Please et al. (2024) [15] for the surgical training study.

The VR hardware employed included a range from Oculus Quest 2 and HTC Vive Pro (used in both studies) to simple cardboard headset viewers paired with a smartphone (used in the surgical training study only), to ensure wide accessibility and compatibility with various levels of technological infrastructure available to learners. Both studies were delivered at the ACE VR visualization lab at the Infectious Diseases Institute (IDI) of Makerere University in Kampala [18], with extensive collaborations from other groups in both studies.

## 2.6 Defining Virtual Reality categories

VR refers to computer-generated simulations that create immersive, three-dimension environments that users can interact with through specialized hardware. These environments allow users to learn from experiences as they would in real life [11]. These experiences are broadly divided into two main types: 360-degree videos and interactive VR. 360-degree videos, often passive, provide a panoramic view of the environment but traditionally limit user interaction [11]. In contrast, interactive VR provides a dynamic and fully immersive environment where users can interact with virtual elements as they would in the real world. This includes simulations of virtual wards, interactive patients, and medical procedures; offering a deeper level of engagement and practical training [11]. Both types were used in our studies as different pedagogical goals require different technologies. For the COVID-19 IPC study, we enhanced 360-degree videos with embedded interactive assessments, transforming the passive viewing experience into an engaging learning activity. Furthermore, this study included a fully interactive module where learners practiced donning and doffing PPE; the system provided real-time audio cues for incorrect steps and prevented users from advancing until the procedure was performed correctly. In the surgical training study, where the goal was live demonstration to a large audience, 360-degree videos were used with minimal processing to allow live-viewing of content by participants in Uganda. This demonstrates the flexibility of the SomaVR framework model to guide technological adaptation to the course-delivery type and user needs.

## 2.7 SomaVR system architecture

### 2.7.1 Software development description using Unity3D.
For the COVID-19 IPC study a SomaVR system architecture was developed to produce the training system. This architecture uses Unity3D (https://unity.com/), a versatile

game engine capable of creating cross-platform VR applications to design a bespoke 360-degree immersive video training platform [19] (Fig 2). The system architecture comprises five critical components described in detail below:

## 1) User Interaction Layer

The user interaction layer integrates VR hardware interfaces with diverse input mechanisms, including touch controllers and gaze tracking commands and employs a custom Unity Canvas system for interactive element overlay without disrupting immersion.

## 2) Core Unity3D Components

Central to the system's functionality is the Unity3D scene management framework. A 360-degree skybox creates an immersive environmental backdrop, while render texture configuration ensures high-fidelity video playback. The integrated video player component streams educational content directly within the VR environment. An interactive question system allows for strategic embedding of multiple-choice assessments at predetermined video timestamps.

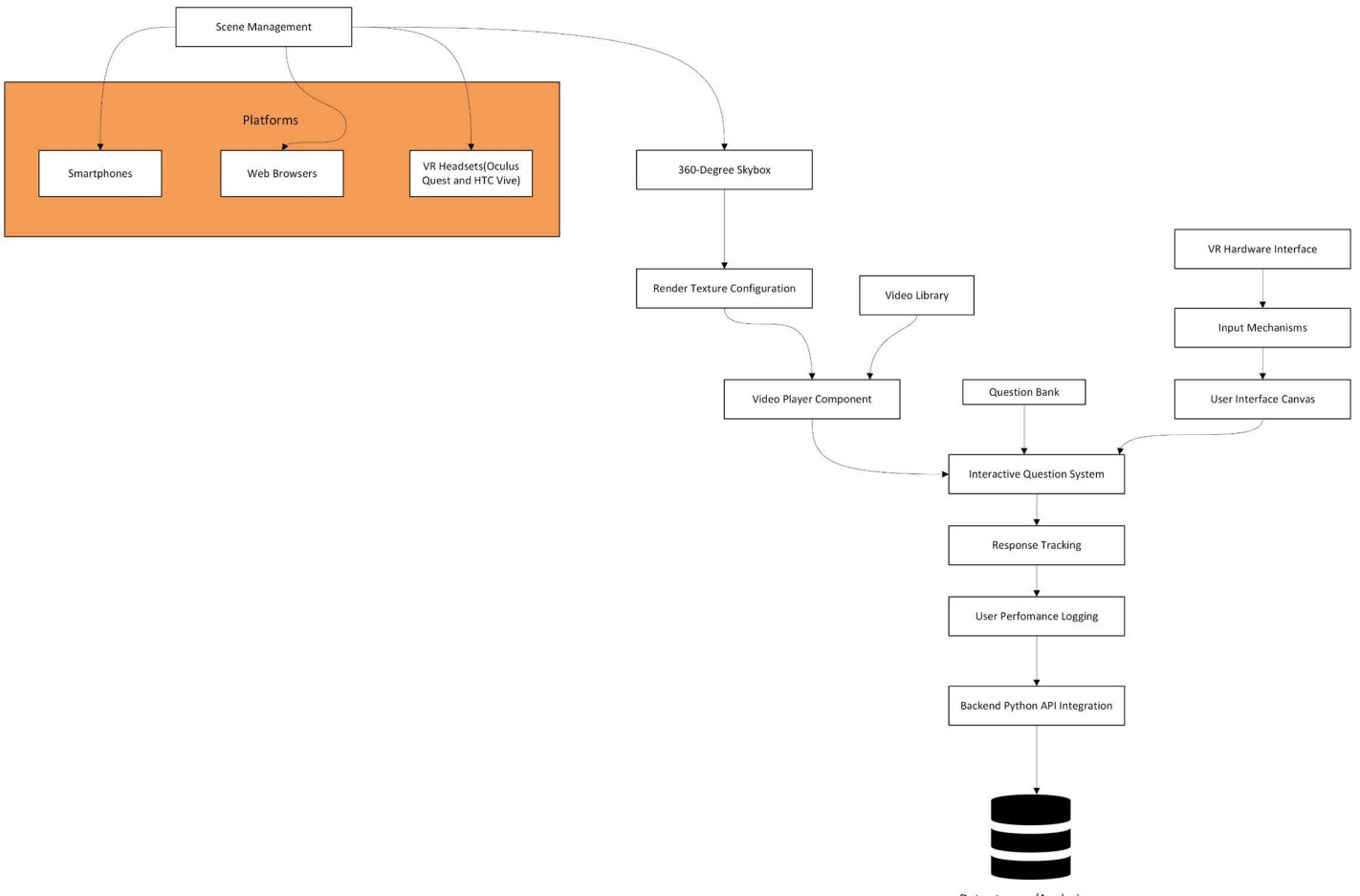

**Fig 2. The SomaVR (used in the COVID-19 IPC study) system architecture: of the 360-degree immersive video training platform developed in Unity3D.** The architecture illustrates the integrated components of the VR training system, demonstrating the flow from user interaction (top) through video content management and interactive assessment systems to backend data tracking and analysis (bottom). Key components include VR hardware interfaces, scene management, video player, interactive question system, and backend API integration, designed to support cross-platform deployment across smartphones, web browsers, and VR headsets.

3) **Data Management**

A robust data management infrastructure tracks user interactions and performance. Response tracking mechanisms capture user selections, response times, and assessment outcomes. These data points are logged and transmitted via a backend API, enabling comprehensive performance analysis and research insights.

4) **Content Management**

The system incorporates a flexible content management approach. A centralized video library stores 360-degree educational content, while a modular question bank allows for dynamic integration of assessment materials. This design facilitates easy content updates and expansion of the training modules.

5) **Platform Support**

Leveraging Unity3D's cross-platform development capabilities, the application supports multiple hardware platforms, including smartphones, web browsers, and various VR headsets ensuring broad accessibility across technological ecosystems.

The code for the platform has been published on GitHub (https://github.com/aceuganda/somavr) and the detailed documentation on how to run it is also provided on the GitHub page.

**2.7.2 Development and deployment.** The development process of the SomaVR system architecture in the COVID-19 IPC study focused on creating an interactive and intuitive learning experience. Each training module was designed to integrate video content with strategically placed interactive assessments. The user interface was designed to be intuitive, allowing learners to engage with content through various input methods without breaking immersion. Comprehensive testing was conducted to ensure consistent performance across supported platforms including Oculus Quest, Smartphones, HTC Vive Pro. User feedback and iterative refinement were integral to the development process, with continuous improvements made based on actual user interactions and performance data. The final application was optimized to meet VR platform distribution standards while maintaining the core educational objectives of the training system.

**2.7.3 Integration with Enduvo.** To enrich the COVID-19 IPC study environment, we integrated Enduvo's (https://enduvo.com/) platform, which allowed for the importation and manipulation of traditional educational materials, such as PowerPoint slides, into VR-compatible formats. This facilitated the creation of interactive learning modules that learners could engage with in a dynamic VR setting, simulating a classroom experience. The course content can be accessed here (https://my.enduvo.com/course/6VuYTzTXY).

## 2.8 Implementation and evaluation studies

The effectiveness of the SomaVR framework was then assessed through the primary studies, each addressing distinct aspects of medical training. While this manuscript focuses on the technical development and implementation of the framework of the SomaVR, we provide a methodological summary of the evaluation studies (Fig 3). Fig 3 provides a brief comparative overview of the study designs and VR delivery (for more detailed descriptions of curriculum design, implementation, and evaluation, see our published studies in Buyego et al. (2022) [7]and Please et al. (2024) [15]).

**2.8.1 Study 1: COVID-19 IPC training.** The first study focused on the feasibility and effectiveness of VR-based training for COVID-19 IPC among healthcare workers in Uganda [7]. The study employed a randomised controlled design comparing VR-based training to traditional classroom instruction. Participants were divided into two groups: one group received traditional classroom training, and the other underwent VR training using both the 360-degree and interactive VR modules of the platform. The participants consisted of healthcare workers from various hospitals across Uganda. The evaluation involved pre- and post-training assessments to measure knowledge retention, procedural accuracy, and confidence in performing IPC tasks. Additionally, feedback was collected to gauge participant satisfaction and the perceived value of VR training. This comparative approach allowed for a detailed assessment of VR training's impact relative to traditional methods.

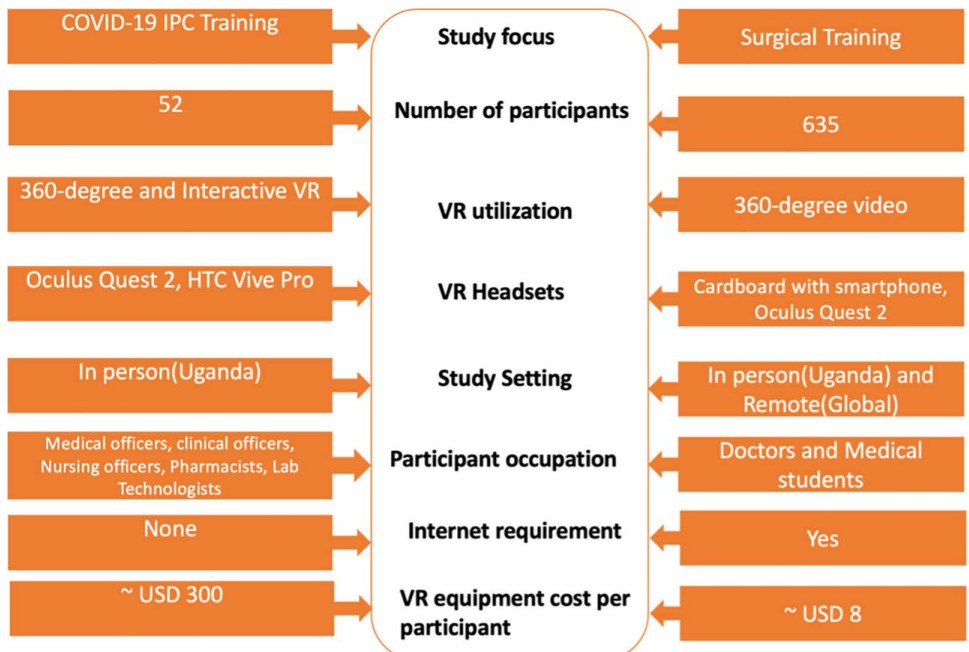

**Fig 3. Comparative overview of the COVID-19 IPC training and surgical training studies: a side-by-side comparison of the two primary studies conducted to evaluate the SomaVR framework.** On the left, the COVID-19 IPC study involved 52 healthcare professionals from Uganda using both 360-degree and interactive VR facilitated by Oculus Quest 2 and HTC Vive Pro headsets. On the right, the surgical training study involved 635 participants, including doctors and medical students, using 360-degree videos accessible through cardboard viewers with smartphones and Oculus Quest 2. This infographic highlights differences in the number of participants, VR technology utilization, headsets used, study settings, and the occupational backgrounds of the participants, illustrating the diverse applications and scalability of the VR training environments.

**2.8.2 Study 2: Surgical training.** The second study explored the potential of VR to enhance surgical skills and knowledge of doctors and medical students, focusing on emergency and essential surgical procedures [15]. Unlike the first study which employed a randomized controlled design, the second study used an observational approach without a control group focusing on accessibility, acceptability, and user experience metrics. This study specifically employed the 360-degree VR, offering a less interactive but still immersive experience compared to the multi-modal approach used in the IPC training. This allowed high volume of procedural recordings with minimal processing, allowing content to be shared live with course participants. Participants included surgical residents and practicing surgeons from multiple LMICs. They underwent VR training on emergency surgical procedures and were evaluated on several key performance metrics, including skill acquisition, accessibility and acceptability, and interest.

### 2.9 Ethics statement

**2.9.1 Ethics approval and consent to participate.** This study is a secondary analysis of previously published data. All data were fully anonymized before access. The two case studies that serve as the basis for this technical paper received independent ethics approval. The software presented does not have access to any information that could identify individual participants during or after data collection. The COVID-IPC study was operational research which received permission from the Ministry of Health and was approved by the Joint Clinical Research Centre Institutional Review Board (Uganda). It was performed in line with the principles and guidelines of the Declaration of Helsinki. Participants were informed about the anonymized use of the training results and provided corresponding written informed consent for participation in the study. The surgical training study received ethical approval by the Social Sciences & Arts, Cross

Schools Research Ethics Committee, University of Sussex (reference number ER/DP254/7). Participants gave informed consent to participate in the study before taking part. Written informed consent for publication was obtained from the participants.

## 3. Results

### 3.1 Immersive 360-degree

The SomaVR platform integrates 360-degree video modules to provide immersive and interactive training for medical professionals. In the COVID-19 IPC project (Fig 4A), healthcare workers are guided through essential procedures such as donning and doffing personal protective equipment (PPE), hand hygiene, and waste management in a virtual hospital setting. Visual cues, interactive prompts, and real-time feedback ensure that learners engage deeply with the material, simulating realistic high-stakes clinical environments. In the surgical skills module (Fig 4B), learners use low-cost cardboard viewers to practice foundational surgical techniques in a simulated operating room. The module focuses on key procedural steps and allows repeated practice in a safe, controlled virtual setting.

### 3.2 Interactive VR in Enduvo

Enduvo delivers theoretical and didactic content in an interactive VR environment, complementing the 360-degree video-based training in SomaVR. It allows educators to create structured lessons with slides, 3D models, and narrated explanations, making it ideal for medical education. In the COVID-IPC study, participants used Enduvo to access instructional materials in a virtual classroom-like setting. Interactive elements, including annotated images, 3D artifacts and embedded quizzes to enhance engagement (Fig 5)

### 3.3 Evaluation of outcomes from prior studies

Fig 6 demonstrates user responses from the surgical training study on the ease of accessing VR content, comparing participants in Kampala and remote settings. The majority of respondents across both groups found the VR content relatively easy to access, with higher ease reported in remote settings. These findings highlight the accessibility of VR training in diverse settings, supporting its scalability in LMIC contexts, as detailed in Please et al. (2024).

Fig 7 presents a comparison of virtual reality training scores between trained and untrained cohorts during the COVID-19 IPC study. The trained cohort exhibited higher performance, with scores clustered in the upper percentiles, while the untrained cohort showed greater variability and lower average scores. These results emphasize the efficacy of VR training in improving learning outcomes, as described in Buyego et al. (2022) [7].

### 3.4 VR equipment cost breakdown

The economic feasibility of deploying the SomaVR platform was assessed by detailing the costs associated with purchasing various types of VR equipment used in the training studies. This cost breakdown provides insights into the financial investments required and the funding sources that supported the implementation (Table 1). It also demonstrates the variability of VR costs depending on educational course size. The costs and usability need to be weighed with other factors, for example reusability and technical experience required. For example, the Oculus and HTC Vive likely have higher longevity than a cardboard viewer which can be more prone to wear & tear, however the Oculus and HTC Vive require more technological familiarity for set-up than the cardboard viewers.

## 4. Discussion

In this study, we present the development of an improvised and affordable VR platform designed to accommodate various educational and training needs specifically in the health-care setting, making it pertinent in low- and middle-income

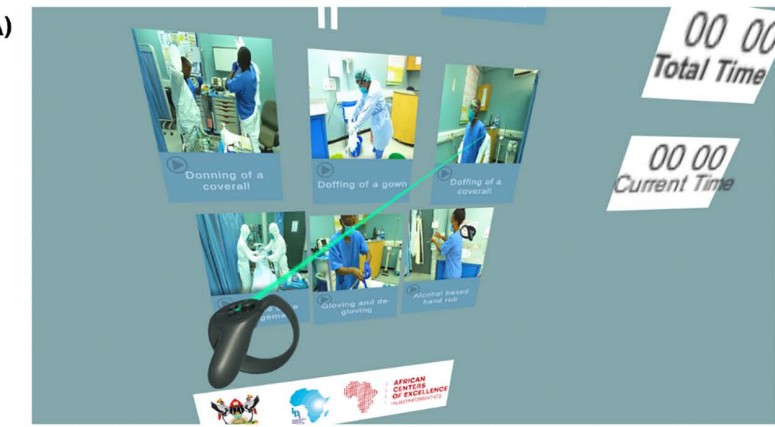

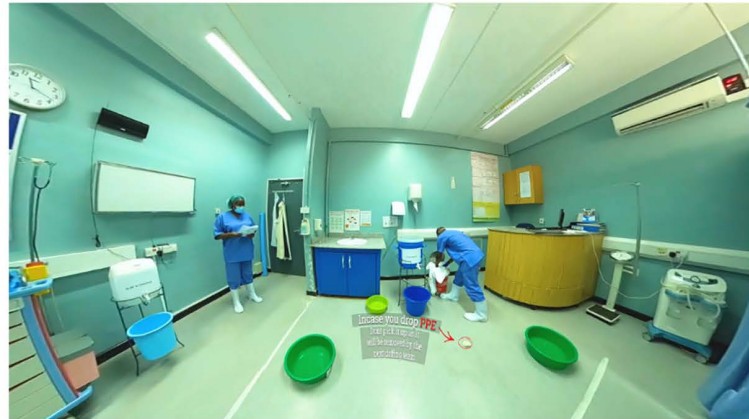

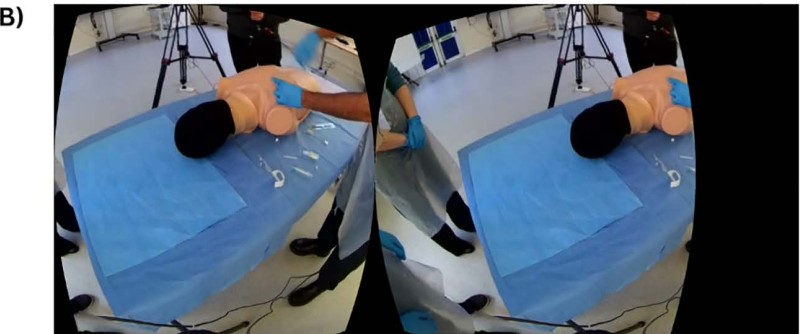

**Fig 4. 360-degree training modules in SomaVR. (A)** IPC module: A virtual hospital environment showcasing essential procedures such as donning and doffing PPE, waste management, and hand hygiene, enhanced with interactive elements and visual guidance. **(B)** Surgical skills module: A simulated operating room designed for learners to practice foundational surgical techniques using VR headsets or cardboard viewers, ensuring repeated and safe practice opportunities in a virtual environment.

countries (LMICs) [20]. The platform was developed to support both online and offline functionalities and was trialed with different types of VR content and hardware, showcasing its versatility and adaptability [7,15]. This inclusivity in design ensures that the platform is not only technically robust but also contextually relevant for LMIC settings where connectivity and technical resources can be limited.

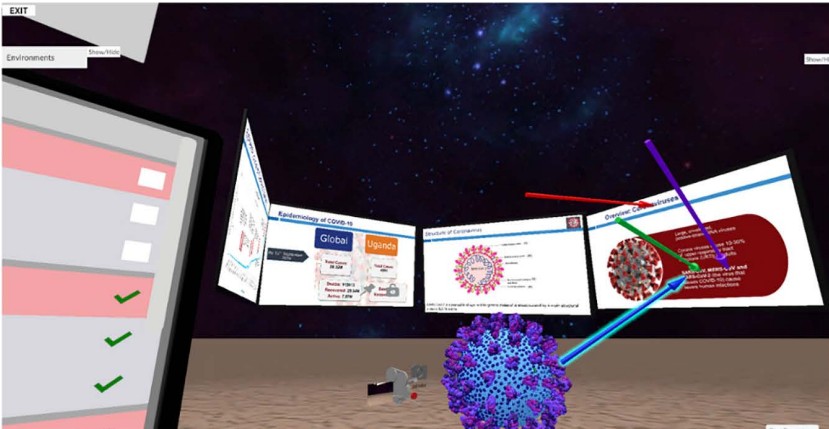

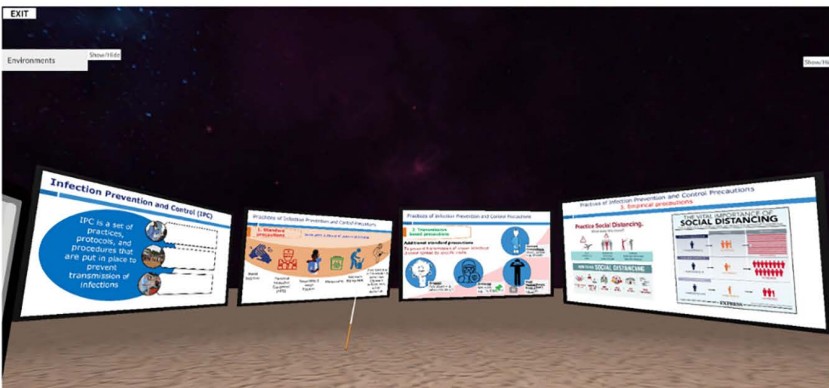

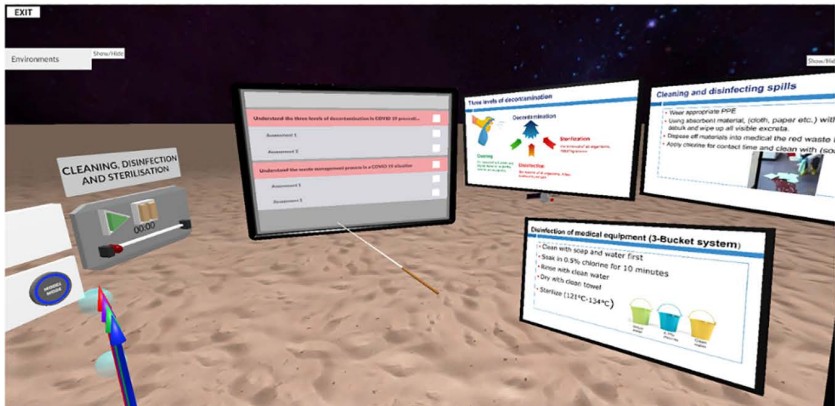

**Fig 5. Enduvo.** A virtual learning environment showcasing learning material in form of traditional slides and 3D models of objects.

The development covered everything from the initial design to the delivery of an immersive VR experience facilitated by the collaborative efforts of multidisciplinary teams. This collaboration included software developers, educational specialists, clinicians, technical support staff and end-users, ensuring a comprehensive approach to delivering engaging and effective educational content. Although assembling multidisciplinary teams incurs significant costs, these investments

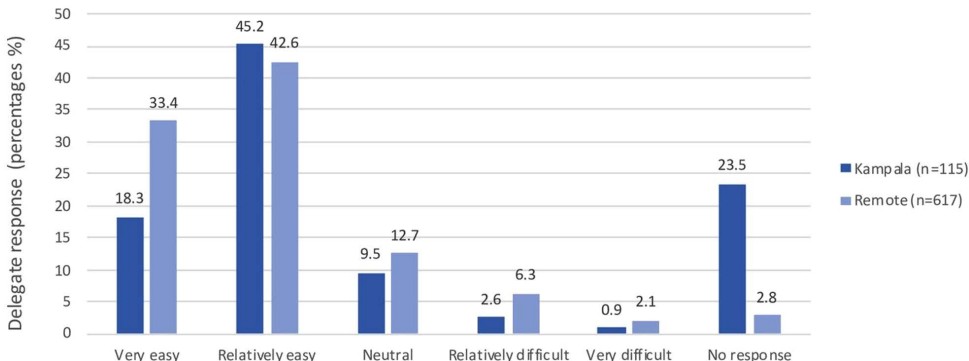

**Fig 6. User-reported ease of accessing VR content during the surgical training study, categorized by participants in Kampala (n = 115) and remote locations (n = 617).** Data shows the majority found accessing the content relatively easy, indicating the adaptability of VR technology across various geographical and infrastructural contexts, validating our cross-platform approach (Please et al., 2024) [15].

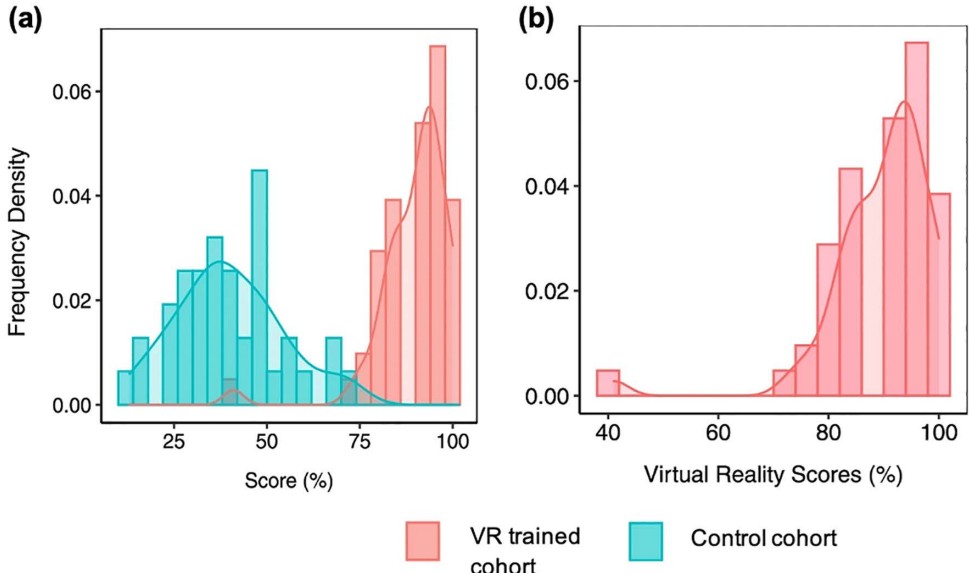

**Fig 7. (a) Virtual reality training scores in the COVID-19 IPC study for VR-trained cohort vs control cohort receiving traditional classroom training (n = 60) (left panel). (b)** Subset of panel (a) showing virtual reality scores (right panel). The VR-trained cohort demonstrated significantly higher and more consistent scores validating that the SomaVR produces effective training systems (Buyego et al., 2022) [7].

are justified by the resultant high-quality medical courses. Such courses have generally been well-received by learners, demonstrating the value of interdisciplinary input in educational content development [12,21].

The inclusion of VR in diverse training contexts has shown measurable benefits, as evidenced by our previous studies. In the surgical training study, participants from both Kampala and remote locations reported that accessing the VR content was relatively easy, with a higher proportion of remote participants finding it very easy. These findings support the scalability of VR technology, even in resource-limited and geographically dispersed settings [15]. Similarly, the COVID-19 IPC study demonstrated the superiority of VR-based training in enhancing knowledge acquisition, with trained cohorts

**Table 1. A comprehensive breakdown of the costs incurred for the VR equipment utilized in the SomaVR training programs. Note: Personnel and internet costs are excluded as they depend on local conditions. In our case, staff were institutional employees and internet was provided via the RENU partnership. For reference, see Section 2.1 for the person-hours and Section 2.4.1 bandwidth requirements.**

| Item | Unit Cost ($) | Quantity | VR Type |
|---|---|---|---|
| Oculus Quest 2 | 299 | 2 | Supports both Interactive and 360-degree videos |
| HTC Vive Pro | 800 | 8 | Supports both Interactive and 360-degree videos |
| Cardboard viewers | 10 | 100 | Only supports 360-degree videos |
| High-end workstation | 2,000 | 8 | Needed if using the HTC Vive Pro |
| 360-degree camera | 400 | 1 | Needed for 360-degree content creation |
| Unity3D License | – | – | Needed for both Interactive and 360-degree videos |
| Enduvo License | – | – | Needed for Interactive VR |

significantly outperforming their untrained counterparts [7]. Together, these results underscore the adaptability, accessibility, and efficacy of VR in meeting varied educational needs across LMIC contexts, validating the technical framework and design principles described in this study.

Despite the noted benefits, the cost of VR equipment remains a significant drawback, consistent with findings from other studies [12]. To mitigate that constraint, we utilized cardboard viewers for delivering 360-degree video content, a cost-effective solution that, although less immersive than interactive VR, substantially reduces the initial financial burden and still enhances content delivery. The affordability and ease of sourcing cardboard viewers presented an effective compromise between cost and functionality [22,23]. In the surgical study cardboard viewers were used for the majority of the study, with an option for a short experience of the material using Oculus Quest 2; allowing for a frugal delivery paired with an opportunity for participants to also experience high-definition technology.

Interactive VR, while initially more expensive due to sophisticated equipment requirements, offers significant long-term benefits through the reusability of its educational content. Well-crafted modules can be repeatedly utilized and widely distributed, optimizing the cost-benefit ratio over time. Moreover, the costs of VR hardware and software have consistently declined, making the technology increasingly accessible [24], For example, the HTC Vive Pro, priced at approximately $1,099 at launch in 2018 [25], dropped to $300 in 2024. During the 2022 COVID-19 IPC study, a setup using the HTC Vive Pro and a high-end workstation cost an estimated $3000. In contrast, the Oculus Quest 2 (64 GB storage), launched in 2020 at $299 [26], eliminated the need for a standalone station due to its wireless, lightweight design. By 2024, its price has further decreased to $249 (128 GB) [27], exemplifying the rapid evolution and affordability of VR hardware.

To mitigate common user experience issues associated with VR, such as motion sickness, nausea, dizziness, and fatigue, which have been widely reported [12], our courses were intentionally designed to be short, each under five minutes. This consideration helped ensure that participants did not suffer from the typical discomforts associated with longer exposure to VR environments.

## 4.1 Limitations

Despite the advantages of using Unity3D, recognized for its extensive support and robust capabilities for VR development, we encountered significant challenges due to the scarcity of local Unity3D developers in Uganda. Most computing graduates in this region specialize in more traditional and immediately lucrative fields such as web and mobile app development [28]. This specialization trend poses a barrier to the adoption and scaling of VR technologies, crucial for advancing educational and training methodologies in underserved areas. To address these challenges and reduce dependency on expensive and unsustainable outsourcing from high-income countries (HICs) we initiated a capacity-building program at our center. This involved establishing an internship program aimed at training local developers in VR technology, focusing on Unity3D. This initiative not only helped lower the costs associated with the development and maintenance of VR

applications but also fostered a sustainable skill set within the local community, contributing to long-term technological self-sufficiency.

During the surgical training study, we encountered bandwidth limitations when over 30 on-site participants simultaneously streamed 360-degree video content on the first day. This caused buffering issues for local participants although remote participants in other parts of the country were unaffected. Our internet provider RENU promptly increased our bandwidth to 1 Gbps resolving the problem for subsequent days. The key lesson for implementers is to calculate bandwidth needs.

### 4.2 Future work

To enhance the utility and reach of the SomaVR platform, several future directions can be pursued. Expanding the curriculum to include additional medical specialties such as emergency medicine, pediatric care, obstetrics and gynecology, mental health interventions, and specialized surgical techniques would broaden the platform's applicability. Geographic adaptation is another critical area, with plans to localize content for different linguistic and cultural contexts, develop region-specific medical training scenarios, and create modular content that can be easily translated and customized. Establishing regional training hubs with minimal infrastructure requirements would further improve accessibility in resource-limited settings.

The recent launch of Google's Android XR platform with Gemini models offers opportunities to streamline VR content creation [29]. By leveraging generative AI, immersive and adaptive 3D environments can be developed more efficiently, reducing costs and enhancing scalability for platforms like SomaVR, particularly in resource-limited settings.

Developing lightweight versions of the platform for devices with limited computational capabilities and enhancing offline functionality for areas with restricted internet connectivity are essential to ensuring the platform remains accessible and scalable in diverse settings. These advancements would collectively support the long-term sustainability and global impact of the platform. Additionally, addressing common user experience issues such as motion sickness and visual fatigue in VR settings remains a priority. Improvements in hardware performance, such as reduced latency and higher frame rates, along with better content design, can significantly mitigate these issues [12]. Future implementations will also address current evaluation gaps by incorporating longer follow-up periods and utilizing standardized evaluation frameworks like User Experience Questionnaire (UEQ) and Unified Theory of Acceptance and Use of Technology (UTAUT) for cross-study comparisons.

## 5. Conclusion

This study presents a comprehensive framework for developing and implementing VR-based medical training platforms in resource-limited settings. Through the detailed documentation of SomaVR's development process, from technical architecture to content creation workflows, we provide a replicable model for institutions seeking to leverage VR technology in medical education. The platform's design principles of offline functionality, cross-platform compatibility, and strategic cost management demonstrate that sophisticated VR solutions can be successfully implemented in LMIC settings.

Our approach to building local development capacity through targeted training programs addresses the crucial aspect of long-term sustainability, reducing dependence on external expertise. The platform's successful validation across different medical training contexts – from infection prevention to surgical skills – confirms the versatility and adaptability of our technical framework. As VR technology continues to evolve and become more accessible, the documented approach and architecture presented here offer a foundational blueprint for future implementations, particularly in resource-constrained environments. This work contributes to the broader goal of democratizing medical education by providing practical guidance for developing contextualized, sustainable VR solutions.

### Acknowledgments

We gratefully acknowledge software and technical support from Enduvo, Inc. (an official ACE partner), and Humulo, Inc.

## Author contributions

**Conceptualization:** Mike Nsubuga, Daudi Jjingo.

**Data curation:** Henry Mutegeki.

**Funding acquisition:** Daudi Jjingo.

**Investigation:** Mike Nsubuga, Grace Kebirungi, Helen Please, Paul Buyego, Jag Dhanda.

**Methodology:** Mike Nsubuga.

**Project administration:** Daudi Jjingo.

**Resources:** Rodgers Kimera, Phil Cruz, Meghan McCarthy, Darrell Hurt, Maria Y. Giovanni, Christopher Whalen, Michael Tartakovsky.

**Software:** Mike Nsubuga.

**Supervision:** Grace Kebirungi.

**Writing – original draft:** Mike Nsubuga.

**Writing – review & editing:** Grace Kebirungi, Helen Please, Paul Buyego, Henry Mutegeki, Rodgers Kimera, Jag Dhanda, Phil Cruz, Meghan McCarthy, Darrell Hurt, Maria Y. Giovanni, Christopher Whalen, Michael Tartakovsky, Daudi Jjingo.

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
