## [Decision Letter · Decision Letter 0]

15 Jun 2025

Response to Reviewers
Revised Manuscript with Track Changes
Manuscript
**Journal Requirements:**

1. Your current Financial Disclosure states, “This work was supported through funding from the Makerere University Research Innovation Fund, and in part through the U.S. National Institute of Allergy and Infectious Diseases, as part of the African Centers of Excellence in Bioinformatics and Data Intensive Sciences (ACE) program (https://ace.niaid.nih.gov), an international and public-private collaboration facilitated by the Foundation for the NIH. We gratefully acknowledge software and technical support from Enduvo, Inc. (an official ACE partner), and Humulo, Inc.”. However, your funding information on the submission form indicates that you received funding from “Makerere University”. Please indicate by return email the full and correct funding information for your study and confirm the order in which funding contributions should appear. Please be sure to indicate whether the funders played any role in the study design, data collection and analysis, decision to publish, or preparation of the manuscript.

2. We ask that a manuscript source file is provided at Revision. Please upload your manuscript file as a .doc, .docx, .rtf or .tex.

3. Please provide separate figure files in .tif or .eps format.

4. Please upload a copy of Figure 1, 2, 3, 4, 5, 6, 7 which you refer to in your text on page 8, 10, 12, 14, 15, 16, 17. Or, if the figure is no longer to be included as part of the submission please remove all reference to it within the text.

**Additional Editor Comments (if provided):**
**Reviewers' Comments:**

**Comments to the Author**

1. Does this manuscript meet PLOS Digital Health’s publication criteria?

Reviewer #1: No

Reviewer #2: Yes

2. Has the statistical analysis been performed appropriately and rigorously?

Reviewer #1: No

Reviewer #2: N/A

3. Have the authors made all data underlying the findings in their manuscript fully available (please refer to the Data Availability Statement at the start of the manuscript PDF file)?

Reviewer #1: Yes

Reviewer #2: Yes

4. Is the manuscript presented in an intelligible fashion and written in standard English?

Reviewer #1: No

Reviewer #2: Yes

Reviewer #1: This manuscript describes a set of mostly 360-videos that the team made to facilitate medical training. Although use of virtual reality to facilitate simulation-based learning for high-skill professions like medicine is promising, it is not clear what the goal of this manuscript is, how it differs from existing publications, and so little information about the included “studies” is provided that it is impossible to follow what was actually done or what the results are. Specific comments include:

1.The introduction seems to suggest that this manuscript is solely focused on describing the VR system itself, but (as the authors point out) there are already several other papers published that detail the system and how it works. How is this manuscript different from these? What are its goals relative to these other papers?

2.Basic, essential information about the two “studies” that the authors refer to is missing, such that the rigor of this work cannot be evaluated. For example, the number of participants in each study, eligibility criteria, and basic demographics are not reported. There is no information about what outcomes were assessed in either study or how, or whether the methods used to assess those outcomes were valid. There is no information about whether participants were randomized, and the two studies appear to use different methods without explanation. The ethics subsection says consent was obtained from all participants but then says “As this study involved the secondary analysis of this already published and anonymized data, no further ethics approval or consent was necessary.” Which is it?

3.Similarly, the results are impossible to follow. Of the two results presented, it is not clear what either is actually evaluating. For example, Fig. 6 purports to show “VR training scores,” but training on what? What is this assessing? The other result focusing on the “ease of use” of the VR seems of limited import. No other results discussed in the Methods section (e.g., feasibility, knowledge retention, satisfaction, etc) are reported.

4.Although most of the manuscript makes it sound like one system or one consistent set of experiences, the use of very different hardware systems with different properties to expose users to different experiences for different applications makes it very difficult if not impossible to understand what is actually being evaluated in the study, and so, what the results mean. Google Cardboard is very different than HTC Vive systems, which are in turn very different than Oculus Quests.

5.It is not clear whether any of the content involved in the system was truly interactive, as the only parts described in detail focus on 360 video. Relying entirely on 360 video has important limitations as a training tool because it does not enable interaction. Interaction is essential for capturing the benefits of simulation-based learning.

6.The manuscript frequently uses excessively venerating phrases like “seamless viewing experience” and that “experiences were meticulously crafted,” without explaining how, which makes the manuscript read like an advertisement for the system rather than an evaluation.

Reviewer #2: plos Digital health D-25-00093

SomaVR: low cost VR platform

General:

This is a well-scoped, well-written report on valuable research programs. The findings have substantial relevance to health care training in LMICs. They are well-suited to publication in PLOS Digital Health.

Example of relevance: Our lab has explored AR as a means to test-drive deployments of new gear to remote locations, and the findings in this paper are valuable to our efforts.

The paper, in my opinion, requires very little work to be ready for publication. Below I list a few odds and ends. The comments about section 3.3 are most important.

Comment for PLOS Digital Health: Line numbers make review much easier. Is there a way to require these in the template for submissions?

Comments for authors:

"as follow-up to Financial Disclosure": fill out if required (currently empty)

"recent review by Mergen et al" add reference numbers

"SomaVR's potential to improve ...": also: long-distance knowledge exchange

Section 2.2: Perhaps you could add brief definitions of VR and AR, since this is not clear to all readers.

2.3, 2.6: Great mix of expertise on team

2.7.1 (1): missing "and"

2.7.1: Perhaps you could include examples of videos in an online S.I. (if you have not already)

2.7.1: Please include github link here as well as (or instead of) in "Data and Code Availability".

Study 2: Please add a sentence to clarify that there was no test cohort vs control cohort structure to this study, but that it was observational. (relevant since study 1 did have test-control cohort structure).

3.1: "allows repeated practice in a safe...": I imagine this would be valuable to have before tangling with a Marburg outbreak.

3.3: A few things:

Fig 6 and 7 appear to be switched. I suggest giving results in the same order as the experiments are described in Section 2.8. This is clearer for the reader.

Current Fig 6: It would help the reader if you clarified that panel (a) is a subset of panel (b) (perhaps switch the 2 panels for a more natural left-to-right reading).

Current fig 6 caption: "the untrained cohort" is confusing, since they did receive traditional classroom training. Perhaps refer to this cohort as "control cohort".

Good limitations section

Good conclusion: "Our approach etc"

Thank you for an interesting paper!

**Do you want your identity to be public for this peer review?** For information about this choice, including consent withdrawal, please see our Privacy Policy

Reviewer #1: No

Reviewer #2: No

**Figure resubmission:****Reproducibility:** To enhance the reproducibility of your results, we recommend that authors of applicable studies deposit laboratory protocols in protocols.io, where a protocol can be assigned its own identifier (DOI) such that it can be cited independently in the future. Additionally, PLOS ONE offers an option to publish peer-reviewed clinical study protocols. Read more information on sharing protocols at https://plos.org/protocols?utm_medium=editorial-email&utm_source=authorletters&utm_campaign=protocols

---

## [Decision Letter · Decision Letter 1]

28 Aug 2025

Response to Reviewers
Revised Manuscript with Track Changes
Manuscript
**Journal Requirements:**

1. Please amend your online detailed Financial Disclosure statement. This is published with the article. It must therefore be completed in full sentences and contain the exact wording you wish to be published.

a) State the initials, alongside each funding source, of each author to receive each grant, if applicable. For example: "This work was supported by the National Institutes of Health (####### to AM; ###### to CJ) and the National Science Foundation (###### to AM)."

For more information, please go to our submission guidelines:

https://journals.plos.org/digitalhealth/s/submission-guidelines#loc-financial-disclosure-statement

2. Please ensure that the funders and grant numbers match between the Financial Disclosure field and the Funding Information tab in your submission form. Note that the funders must be provided in the same order in both places as well.

3. We have amended your Competing Interest statement to comply with journal style. We kindly ask that you double check the statement and let us know if anything is incorrect.

4. Some material included in your submission may be copyrighted. According to PLOS’s copyright policy, authors who use figures or other material (e.g., graphics, clipart, maps) from another author or copyright holder must demonstrate or obtain permission to publish this material under the Creative Commons Attribution 4.0 International (CC BY 4.0) License used by PLOS journals. Please closely review the details of PLOS’s copyright requirements here: PLOS Licenses and Copyright. If you need to request permissions from a copyright holder, you may use PLOS's Copyright Content Permission form.

Potential Copyright Issues:

Figures 1 and 2: Please confirm whether you drew the images / clip-art within the figure panels by hand. If you did not draw the images, please provide (a) a link to the source of the images or icons and their license / terms of use; or (b) written permission from the copyright holder to publish the images or icons under our CC-BY 4.0 license. Alternatively, you may replace the images with open source alternatives. See these open source resources you may use to replace images / clip-art:

- https://openclipart.org/

**Additional Editor Comments (if provided):**
**Reviewers' Comments:**

**Comments to the Author**

Reviewer #2: All comments have been addressed

Reviewer #3: All comments have been addressed

Reviewer #4: (No Response)

publication criteria?

Reviewer #2: Yes

Reviewer #3: Partly

Reviewer #4: Partly

3. Has the statistical analysis been performed appropriately and rigorously?

Reviewer #2: N/A

Reviewer #3: No

Reviewer #4: N/A

4. Have the authors made all data underlying the findings in their manuscript fully available (please refer to the Data Availability Statement at the start of the manuscript PDF file)?

Reviewer #2: Yes

Reviewer #3: No

Reviewer #4: Yes

5. Is the manuscript presented in an intelligible fashion and written in standard English?

Reviewer #2: Yes

Reviewer #3: Yes

Reviewer #4: Yes

Reviewer #2: All good!

Reviewer #3: This manuscript presents SomaVR, an affordable and modular virtual reality (VR) training platform, developed specifically for medical education in low- and middle-income countries (LMICs). The work is timely and relevant, responding to persistent gaps in training infrastructure and workforce development across resource-limited settings. The integration of 360-degree video and interactive VR modules, combined with open-source code availability and capacity-building strategies, represents a compelling and replicable approach.

The authors are commended for presenting not only the technical development but also two applied case studies (COVID-19 infection prevention and surgical training). These add practical validation and highlight the potential impact of the SomaVR framework. The manuscript is well-written, logically organized, and clearly grounded in the global health context.

However, there are areas requiring minor revision to improve methodological clarity, strengthen the evaluation component, and streamline the presentation. Below are detailed comments to guide your revision:

1. Significance and Originality

The concept of a low-cost, scalable VR framework for LMICs is original and well justified.

The approach stands out by combining hardware flexibility, offline delivery capability, and a sustainable local developer model.

Suggestion: Consider briefly comparing SomaVR to other LMIC-focused VR platforms, if any exist, to better position its novelty.

2. Methodological Rigor

The description of software architecture and hardware setup is very detailed and helpful.

However, the evaluation design lacks methodological transparency. Please provide:

Clarification on how participants were selected and grouped (randomized or not).

Descriptions of assessment tools used in pre-/post-testing.

Statistical analysis details: were significance tests performed? Were effect sizes calculated?

Actionable Recommendation:

Add a table summarizing key features of the two evaluation studies (sample size, duration, modality, metrics, statistical tests, key results).

3. Results and Data Presentation

The training outcomes are presented clearly, and the figures illustrate core findings effectively.

However, Figures 6 and 7 need better labeling and contextual explanation:

Define “trained” and “untrained” cohorts directly in figure captions.

Indicate sample sizes, statistical significance (if any), and data collection methods.

Recommendation: Consider adding a brief comparative summary of knowledge acquisition scores or engagement indicators across the two cohorts.

4. Technical Description of SomaVR

The software architecture section is detailed and informative. The use of Unity3D is appropriate, and the breakdown of its components is helpful.

The Enduvo integration is a strength, as it supports blended didactic and procedural content.

Minor Suggestion: Some subsections (2.7.1) could be more concise to avoid redundancy. For example, interaction mechanisms and platform support are discussed in multiple places with overlapping phrasing.

5. Discussion and Implications

The Discussion section is thoughtful, especially regarding costs, accessibility, and motion sickness mitigation.

The authors rightly emphasize the challenge of VR developer availability in LMICs and their solution via capacity-building is commendable.

Enhancement Suggestions:

Discuss possible limitations in learner evaluation (e.g., bias, lack of long-term follow-up).

In the Future Work section, consider listing more concrete next steps (e.g., number of modules to be created, training hubs planned).

6. Clarity and Structure

Overall writing quality is strong. The manuscript is well-organized and flows logically.

That said, the Methods and Discussion sections are overly long. Some compression would improve readability.

Language edits:

Replace redundant phrasing (“immersive and interactive” is used excessively).

Ensure consistent terminology for technologies (e.g., use “360-degree video” consistently).

7. Ethics, Open Science and Data Sharing

Ethical considerations are appropriately addressed and conform to best practices.

Open sharing of the SomaVR code on GitHub is commendable and aligns with FAIR principles.

Improvement Suggestion:

Consider archiving the GitHub repository using Zenodo or a similar service to obtain a permanent DOI.

8. Figures, Tables, and Supplementary Materials

Figures 1–5 are informative. Figure 1 (SomaVR framework) could benefit from clearer labeling of each module.

Table 1 (Cost breakdown) is relevant. Adding a “Reuse/Usability” or “Learning Curve” column would increase practical value.

Figures 6 and 7 are important but currently not fully self-explanatory. Improve captions and add statistical context if available.

Conclusion

This manuscript offers a valuable contribution to global digital health and medical education. With minor revisions focused on methodological transparency, concise writing, and clarity in results presentation, this work will be of high interest to the readership of PLOS Digital Health.

Reviewer #4: Summary

This paper presents a framework for implementing low-cost VR-based education infrastructure in low- and middle-income countries (LMICs), with case studies focused on COVID-19 infection prevention and surgical training. The study offers valuable insights into the feasibility of using existing VR technologies in resource-constrained settings.

As a computer scientist, I approached the paper with particular interest in the technical and organizational aspects. From this perspective, I find the overall concept and contribution important and timely. However, the technical components, including system setup, infrastructure requirements, and cost breakdowns, require further clarification and refinement to fully support replication and implementation.

Major Comments

1.Clarity and Structure of Methods Section

The methodology section would benefit from reorganization. I recommend presenting the content from abstract to concrete: start with administrative and organizational aspects, then progress to infrastructure, hardware, systems, and finally evaluation. This structure would improve clarity and flow, especially for readers seeking practical implementation guidance.

2.Inconsistent Terminology on Content Delivery

There is a lack of clarity regarding whether the training content was delivered live or asynchronously. Section 2.1 mentions "live content," while Section 2.4 describes the surgical training as using "entirely online content." If the content was live, was it also recorded? If not, why was asynchronous access not provided?

3.Missing Technical Specifications

For a technically oriented audience, the paper lacks essential details on system requirements. Specifically:

oWhat are the average file sizes for downloadable content in Study 1?

oWhat is the minimum bandwidth needed to support live streaming in Study 2?

These specifications are necessary for readers aiming to replicate the project in LMIC settings.

4.Team Size and Resources

Please provide more information on the deployment teams:

oHow many people were involved?

oWhat was the estimated time commitment (e.g., person-hours) per team?

This would help assess the scalability and resource implications of the proposed framework.

5.Cost Reporting

While the breakdown of VR-related costs is helpful, the paper does not include estimates for internet access or personnel costs. These are significant components in LMIC implementations and should be addressed, at least at a high level.

Minor Comments

•Chapter 2.1 Scope and Focus

The current title (“Hardware“) is overly broad and does not clearly reflect the structure of the paper. Consider emphasizing the organizational implementation first, and relocating the hardware-focused discussion to a later section.

•Figure 1 Legend

The meaning of the colors used in Figure 1 is not explained. Please include a legend or clarify this in the figure caption.

•Use of Standardized Evaluation Tools

For future work, I recommend employing established tools to evaluate user experience and technology adoption, such as:

oUser Experience Questionnaire (UEQ-S)

oUnified Theory of Acceptance and Use of Technology (UTAUT)

Their use would strengthen the evaluation component and allow for cross-study comparisons.

Conclusion

The paper addresses an important topic with strong relevance to global health and education. However, in its current form, the technical and methodological sections require significant improvement for the work to be replicable and broadly impactful. I encourage the authors to revise with attention to the comments above, especially those related to clarity, technical detail, and completeness of cost and deployment data.

**Do you want your identity to be public for this peer review?** For information about this choice, including consent withdrawal, please see our Privacy Policy

Reviewer #2: No

Reviewer #3: No

Reviewer #4: No

**Figure resubmission:**

**Reproducibility:** To enhance the reproducibility of your results, we recommend that authors of applicable studies deposit laboratory protocols in protocols.io, where a protocol can be assigned its own identifier (DOI) such that it can be cited independently in the future. Additionally, PLOS ONE offers an option to publish peer-reviewed clinical study protocols. Read more information on sharing protocols at https://plos.org/protocols?utm_medium=editorial-email&utm_source=authorletters&utm_campaign=protocols

---

## [Decision Letter · Decision Letter 2]

4 Feb 2026

SomaVR: A low-cost virtual reality platform and implementation framework for medical education in resource-limited settings

PDIG-D-25-00093R2

Dear Mr Nsubuga,

We are pleased to inform you that your manuscript 'SomaVR: A low-cost virtual reality platform and implementation framework for medical education in resource-limited settings' has been provisionally accepted for publication in PLOS Digital Health.

Best regards,

Haleh Ayatollahi

Section Editor

PLOS Digital Health

**Additional Editor Comments (if provided):**

**Reviewer Comments (if any, and for reference):**

Reviewer's Responses to Questions

**Comments to the Author**

Reviewer #2: All comments have been addressed

Reviewer #3: All comments have been addressed

publication criteria?

Reviewer #2: Yes

Reviewer #3: Yes

3. Has the statistical analysis been performed appropriately and rigorously?

Reviewer #2: Yes

Reviewer #3: Yes

4. Have the authors made all data underlying the findings in their manuscript fully available (please refer to the Data Availability Statement at the start of the manuscript PDF file)?

Reviewer #2: Yes

Reviewer #3: Yes

5. Is the manuscript presented in an intelligible fashion and written in standard English?

Reviewer #2: Yes

Reviewer #3: Yes

Reviewer #2: I look forward to seeing this paper published. It advances the literature on a valuable topic.

Reviewer #3: The revised manuscript “SomaVR: A low-cost virtual reality platform and implementation framework for medical education in resource-limited settings” is now clear, methodologically sound, and well written. The authors have effectively addressed all previous reviewer comments.

The study presents an innovative and relevant contribution by developing an open-source, low-cost VR platform for medical training in low- and middle-income countries. The methodology is transparent, the data support the conclusions, and the statistical analyses are appropriate for the study’s objectives.

The organization, writing quality, and clarity have improved substantially. Figures and tables now clearly convey the results, and the open availability of the code and materials meets PLOS Digital Health’s standards for reproducibility and open science.

Minor suggestions for future work include:

Expanding validation using standardized usability tools (e.g., UEQ-S, UTAUT).

Including long-term follow-up or benchmarking with similar VR systems.

Overall, this is a technically rigorous, ethically compliant, and impactful paper that contributes meaningfully to global digital health education.

**Do you want your identity to be public for this peer review?** For information about this choice, including consent withdrawal, please see our Privacy Policy

Reviewer #2: No

Reviewer #3: No
